# Measuring remote working skills: Scale development and validation study

**Serap Benligiray**[1]☯**, Abdullah Y. Güngör**[2]☯*, **İlkay Akbaş**[3]☯

**1** Department of Business Administration, Anadolu University, Eskişehir, Türkiye, **2** Department of Business Administration, Atatürk University, Erzurum, Türkiye, **3** Department of Business Administration, Beykoz University, İstanbul, Türkiye

☯ These authors contributed equally to this work.
* yigit.gungor@atauni.edu.tr

**Data Availability Statement:** All relevant data are within the paper and its Supporting information files.

**Funding:** The authors received no specific funding for this work.

## Abstract

Remote work, one of the most significant working arrangements of today, requires certain employee skills. Although there are some hints, there is not much information in the literature on this subject. This study aims to identify the skills required for productive remote working activities and to develop a scale for measuring these skills. For this purpose, a thorough review of the literature, consultation with experts, and analysis of data obtained from four samples with remote working experience were all conducted. Within this context, item generation and content validation, initial factor structure analysis, and factor structure confirmation and construct validity examination were performed. Consequently, the Remote Working Skills Scale was developed, which has 36 items and five dimensions (cybersecurity, problem-solving, time management, verbal communication, and written communication).

## Introduction

The concept of remote working has gradually gained popularity since the 1970s [1], when the first examples can be found [2–5]. Especially since 2020, the pandemic restrictions have forced a large number of people to work from home [6–8]. This obligation has made some employees and organizations realize that the flexible hybrid labor force model has brought about significant benefits and that it is possible for people to work outside of traditional office spaces. Within this context, it is possible to argue that the pandemic is the driving force behind the remote working revolution, which has long been expected but has not been able to be realized on a large scale. The persistence of remote working following the pandemic has been made possible by digitalization, advanced communication and cloud technologies, and it is anticipated that in the coming years, it will become a part of the new and better normal [9, 10]. According to the State of Remote Work 2021 Report of Owl Labs, today, 16% of the world's companies are building the future through 100% remote working [11].

Both employees and organizations may benefit from remote working in various ways. However, the remote working model faces some significant difficulties in practice because it differs significantly from the traditional working models. According to a relevant study, the following 4 elements were found to be effective in the successful implementation of remote working: The individual (personal traits and skills), the job (the nature of the job, technology, etc.), the

**Competing interests:** The authors have declared that no competing interests exist.

organization (strategy, culture, etc.), and home-family [12]. However, a different study found that a variety of factors, including organizational culture, administration, the nature of the job, and skills, have an impact on remote working [13]. Therefore, for a productive remote working environment, various elements should be provided on an organizational, administrative, and employee basis [14]. Organizations should use a variety of strategies to address this situation. These strategies include determining and improving remote workers' skills and offering relevant training programs [15].

Some previous studies [15–22] have shown that certain employee skills are important for the efficient implementation of a remote working model because these skills have a significant impact on employees' performances [23]. Additionally, the costs associated with discord and a lack of skills impact individuals, companies, and the overall economy [24]. As a result, it becomes more difficult to determine the employee skills necessary for remote working, which is currently one of the most important working models.

Even though there are some clues about the employee skills needed for an efficient remote working environment, the relevant literature contains limited content and studies. Various scale development studies have been conducted in the literature [25–27] with the aim of making remote working more successful. However, once these studies were examined, it was seen that the scales were not skill-oriented. For example, in a scale study developed to measure digital competencies [26], although the scale developed by the authors includes certain skills for remote working, it is actually a competency-focused [28] scale with a broader scope. Therefore, it can be stated that since it is a scale that also includes behaviors and attitudes, it is not a scale that specifically measures only skills. In addition, scales have been developed to measure well-being in remote work [27]and attitudes towards remote work [25]. Again, it can be stated that these scales are not skill-oriented. Although these scales are important measurement tools for the success of remote working, they do not measure the remote working skills that the authors focus on in this study. From this viewpoint, it was deemed necessary to identify the skills required for productive remote working activities, and to develop a scale regarding these skills.

The theoretical background of this study is based on Resource-based Theory [29] and Human Capital Theory [30], which considers skills as a valuable resource in the success of organizations. In this context, it has been demonstrated that the developed RWSS is an important tool in the success of the remote working model. Determination of remote working skills was carried out through three basic research questions developed by the authors. Research question one (RQ1): What are the factors required for businesses to be successful in remote work, and which of these are related to employee skills? Research question two (RQ2): What are the factors required for employees to be successful in remote work, and which of these are related to skills? Research question three (RQ3): What are remote working skills? After that, the keywords for each research question were determined and grouped. As a result of literature review and analysis, five fundamental remote work skills dimensions were determined. Subsequently, in line with the scale development stages, item generation and content validation, initial factor structure examination, and factor structure confirmation and construct validity examination were performed. Consequently, a Remote Working Skills Scale was developed, which contains 36 items and five dimensions.

## Remote working

There are numerous terms that have comparable or similar meanings to remote working in the literature, such as telework, telecommuting, virtual work, mobile work, homeworking, working from home, distributed work, working from anywhere, etc. The abundance of these terms also show how far remote working has progressed. For instance, a study [31] categorized

remote working into generations and evaluated its evolutions. The first generation was the home office, which predominated in the 1970s and 1980s; the second generation was the mobile office in the 1990s; and the third generation was the virtual office, popularized in the 2000s and after. In the first generation, the work is basically done at home by full-time employees of an organization through computers and telephones. In the second generation, part-time employees of an organization can work from anywhere through laptops and cell phones. Finally, in the third generation, there is a professional relationship between the organization and the employees, which is established in any manner, and the work is done anywhere through tablets, computers, or smartphones.

There are several definitions of remote working in the literature. Remote working is "the practice of an employee working at their home, or in some other place that is not an organization's usual place of business" [32]; "an arrangement between employee and employer where the employee's work is performed remotely outside the employer's premises thanks to the aid of information and communication technologies" [33]. Di Martino and Wirth [34]define remote working as "a flexible work arrangement whereby workers work in locations, remote from their central offices or production facilities, the worker has no personal contact with co-workers there but is able to communicate with them using technology". According to Kłopotek [35], remote working refers to "an organizational work that is performed outside the normal organizational confines of space and time". In light of these definitions, remote working can be defined as a way of working in which employees carry out their duties from any location other than the workplace via information and communication technologies.

Remote work has some benefits and drawbacks for both individuals and organizations. Individual benefits include increased productivity [13, 21], performance [36], job satisfaction [20, 37–39], motivation [13, 40], and autonomy [13, 38], establishing a better work-life balance [20], and decreased work-family conflicts [38]. In contrast, remote working has some disadvantages, such as social isolation/solitude [38, 41–43], lack of face-to-face communication [44, 45], lack of motivation [13], the negative impacts of distracting factors at home [45], and work-family conflict [38].

On organizational terms, benefits of remote working include increased commitment [13, 36, 37, 39] and productivity [39], improved retention of qualified employees [13, 40], broader pool of skills [38], decreased fixed costs [38], and maintaining the continuity of work in emergencies [40]. On the other hand, lack of face-to-face communication [13, 40], coordination [13], control and monitoring/surveillance [38, 45], security [13], teamwork [40, 45], and cooperation [20] are among the drawbacks of remote working.

## Remote working skills

A skill is a combination of knowledge, experience, and abilities, which allows its users to exhibit better performance [46] or a state of concluding or achieving a job or a process successfully based on their predisposition and educational background. Along with their technical skills for their jobs, remote employees should be able to use computers and other related equipment and software in order to perform as expected. However, employees are expected to have these skills regardless of whether they work at the office or remotely. This study's main subject is the prominent and essential skills for remote work. In this context, remote working skills are defined as the combination of knowledge, experience, and abilities to fulfill job duties successfully during working from any location other than the workplace via information and communication technologies. The literature review conducted to identify the skills related to remote working revealed the following five dimensions: cybersecurity, problem-solving, time management, verbal communication, and written communication.

**Cybersecurity.** Cybersecurity is the set of measures taken to protect a computer or computer system against unauthorized access or attacks [47], and the collection of tools, policies, security concepts, security safeguards, guidelines, risk management approaches, actions, training, best practices, assurance and technologies which can be used to protect the cyber environment and organization and user's assets [48]. A cybersecurity problem will violate confidentiality, integrity, or availability of information [49]. Therefore, security systems should be built into devices in order to resolve a cybersecurity problem. Cybersecurity experts agree that security depends on individuals rather than technical controls and countermeasures [50]. 50–75% of cybersecurity threats are caused by users' intentional or unintentional misuse of Information Systems (IS) resources. Employee security flaws and violations result in threats against IS and other significant financial and information losses [51, 52]. As a result, computer users are considered one of the weakest links within the IS security chain [53]. The errors caused by poor cybersecurity skills of information technologies (IT) users should also be considered as a factor. This constitutes approximately 72–95% of all cybersecurity threats against institutions. For this reason, computer end-users are one of the weakest links within the cybersecurity chain due to their limited skills [54]. Thus, for today's workers, cybersecurity is now a crucial skill. Since the entire work is performed through information technologies in remote working, cybersecurity skills have become more critical. In terms of cybersecurity skills, the technical knowledge, skills, and experiences pertaining to hardware and software required for preventing malware, personal identifiable information (PII) theft, and work information system (WIS) violations; reducing cyber-attacks, and conducting IS security become crucial [51].

A study was carried out by Carlton et al. [55] to evaluate the cybersecurity skills required for non-IT professionals. The skills identified in that study are preventing leakage of confidential digital information to unauthorized persons, preventing malware through non-secure websites or e-mails, access to non-secure websites, PII theft through e-mail phishing or social networks, and credit card frauds during online shopping, and violations of information system security through external storage devices (USB, Hard Disk, etc.) and password theft. Another study emphasized that personal devices, that do not have any customized firewalls, automatic backup tools, and strong antivirus software programs available in corporate networks, should be used without posing any security threats [56].

**Problem-solving.** According to Chi [57], a problem is a difficult and complex situation that needs to be solved through specific tools to achieve a goal or a conflict that prevents individuals from achieving a goal [58]. The process of overcoming challenging and complex problems or addressing issues is known as problem-solving [59]. This process includes carefully examining an existing problem and finding the best solution for that problem.

Problem-solving is an essential skill for both office workers and remote employees. When these two groups are compared, it may be concluded that remote workers can solve problems by themselves better compared to the others. However, a different study by Bowen and Pennaforte [60] discovered that the need for face-to-face interaction for problem-solving activities is one of the most significant challenges faced in the context of remote working. Remote employees have fewer opportunities to exchange ideas with their colleagues or managers regarding the solution to a problem [61]. As a result of increased physical distance and lack of face-to-face interaction in the remote working model, employees' problem-solving skills have become more critical [62, 63].

Employees with problem-solving skills could streamline intra-organizational bureaucracy and enhance the service period [64]. The studies also discovered a positive relationship between problem-solving skills and job satisfaction [65], performance [66, 67], creativity [65, 68], and innovativeness [69]. Within this context, it may be claimed that employees with problem-solving skills have higher levels of job satisfaction, creativity, innovative behaviors, and

performance. Problem-solving usually requires knowledge and skills that help employees deal with challenging and extraordinary situations. Similar to remote work, cases with higher levels of complexity and uncertainty make problem-solving skills become even more crucial [70]. Remote workers need online problem-solving skills in order to formulate the problem and find strategies to determine the best solution. Problem-solving skills for remote workers include determining the digital needs and resources, choosing the digital tools suitable for pre-determined goals and needs, solving conceptual problems through digital methods, using technologies creatively, solving technical problems, and updating digital competencies [71].

**Time management.**   Several studies have discussed time management as a process [72–74]. This process typically includes setting goals for identifying and fulfilling the needs and listing and planning the tasks to achieve these goals. For example, Claessens et al. [75] defined time management as the collection of behaviors designed to use time efficiently while performing activities to accomplish a particular objective.

Time management is crucial in work life. Studies have found a relationship between time management and performance [76–78], job satisfaction [79, 80], creativity [81, 82], stress [83], anxiety and depression [84], work-family conflict [85], and emotional exhaustion [86]. Within this context, it can be said that employees with better time management skills show better performance and are more creative and satisfied with their jobs. However, employees with insufficient time management skills experience high levels of work-family conflict, stress, anxiety, tension, and emotional exhaustion.

Some previous studies [22, 87–89] have also shown that time management is also one of the most essential skills for remote workers. Additionally, time management skills become even more critical in remote working due to certain factors, such as challenges resulting from working together with individuals from different locations and time zones, the need for individual efforts for work schedules, tasks, and responsibilities, and the importance of threading a thin line between work and family responsibilities. Therefore, remote workers with time management skills should perform their duties within certain time management practices. In this regard, it is important to stress the duties of employees who possess time management skills. According to Nickson and Siddons [61], having a clear understanding of job descriptions improves time management. A remote worker should then accurately calculate the time required for each task. This calculation should be followed by task prioritization. Additionally, the remote worker should identify off-duty time-consuming activities and should avoid these activities because remote working activities are usually performed at home, and there may be more distracting and time-consuming factors compared to the office environment.

**Verbal communication.**   Verbal communication takes place face-to-face, on the phone, or in electronic environments, and via talking. In remote working, employees often use verbal communication when performing a task that requires collective work and/or collaboration, and this communication inevitably takes place in a virtual environment. According to a relevant study, verbal communication in a virtual environment increases performance by enabling the employees to discuss various subjects and instantly inform each other, and it also has a positive impact on the sense of coexistence and awareness [90]. Another study, which aimed to measure the success of virtual communication, discovered that verbal communication plays a role in building up trust among employees, encouraging participation, increasing productivity, ensuring goal congruence, and successfully giving/receiving feedback [91]. Different studies show that verbal communication has a significant place among the skills managers and employees should have to succeed in remote working environments [15, 88, 92, 93].

Verbal communication in virtual environments requires different skills compared to face-to-face communication. A relevant study [94] revealed that remote workers from around the globe make adaptations, such as adjusting their speaking speed, choosing fewer complex

words and sentences, and minimizing their accents in verbal communications to increase clarity. Active listening, proper use of grammar, accurate and plain expression of thoughts/opinions, receiving feedback, giving feedback, effective and correct communication in times of crisis, efficient use of the telephone, active participation in meetings, efficient team communication, conflict resolution, persuasiveness, etc. are all examples of verbal communication skills [95–97].

**Written communication.**   Written communication involves conveying feelings, ideas, impressions, and opinions on any matter in writing within the frame of specific rules. Written communication is one of the most essential skills for all employees, regardless of whether they work in an office or remotely. The studies [15, 88, 92, 93, 98, 99] emphasize that written communication plays a crucial role in remote working and will gradually increase in significance. Remote workers and managers should therefore possess advanced written communication skills.

In remote working, business communication changes, and employees should perform their reading, writing, viewing, and content creation activities more efficiently within a broader media field [91]. For example, a study on virtual communication, which enables staff members to communicate and interact through various digital and electronic media types, found that employees rely more on instant messaging, e-mailing, and other similar text-based communication when communicating with their colleagues [100]. Another study revealed that text-based communication constituted the majority of virtual communication in the workplace [101].

Failures in written communication can result in various problems. For instance, remote working team members make wrong assumptions regarding the goals and targets if they misinterpret the meaning and sense of a written text. In addition, emerging interpersonal issues have been shown to lead to conflicts and negatively impact performance [102]. In a study on remote working [103], a global group of remote workers and a traditional group of office workers were compared, and it was found that the remote working group experienced more misunderstandings and misinterpretations. The researchers attributed this to the lack of written and face-to-face communication. They came to the conclusion that sharing written summaries of teleconferences would have a positive impact on eliminating misunderstandings and misinterpretations.

In order to cooperate through written communication in remote working, employees should use plain and simple language, abide by punctuation and grammar rules, and avoid discriminatory and disturbing expressions. Correct spelling, using the appropriate tools depending on the context (visual, statistical, etc.), using suitable formats for different readers, persuasiveness, etc. are additional aspects of written communication skills [95, 104–107].

## Hypotheses

According to the literature mentioned above, employees should have basic cybersecurity, problem-solving, time management, verbal communication, and written communication skills for efficient remote working. Accordingly, we aim to develop and validate a five-factor Remote Working Skills Scale (RWSS): Hypothesis 1- Factor analyses will reveal five comprehensive Remote Working Skills (RWS) factors that represent cybersecurity, problem-solving, time management, verbal communication, and written communication. Numerous studies show that employees with higher skill levels perform better. In these studies, for instance, it was found that both soft and hard skills have a significant impact on employees' work performance [108, 109]; communication skills increase employees' performance [110]; training and improvement are crucial ways to improve the skill set of an employee, and enhance their

performance [16, 23, 111]. Within this context, the second hypothesis of this study was determined as follows: Hypothesis 2- There is a positive relationship between performance and RWS.

According to Meyer and Allen [112], commitment is a psychological state that has 3 components, which represent a desire (affective commitment), a need (continuance commitment), and the obligation (normative commitment) felt for maintaining employment at an organization. Affective commitment has a positive relationship with performance [113–116]. However, this is not the case for other dimensions. For example, several studies concluded that there is no relationship between continuance commitment and employees' performance [117, 118]. However, in a different study by Meyer et al. [119], a positive correlation between affective commitment and performance and a negative correlation between continuance commitment and performance were found. In light of these findings in the literature, H3 and H4 were determined as follows:

Hypothesis 3- There is a positive relationship between affective commitment and RWS.

Hypothesis 4- There is not any relationship between continuance commitment and RWS.

## Materials and methods

### Study design, sampling, data collection, and analyses

The study design is an exploratory sequential mixed method. Thus, remote working skills were determined with a qualitative phase first, then the remote working skills instrument was built, and finally remote working skills scale was tested. In terms of time, the study design is cross-sectional. Therefore, the data of the study were collected at a single point in time on the specific samples.

The sampling technique of the study is a purposeful sampling technique. In accordance with the research design, 3–10 participants are sufficient for the qualitative phase [120]. The number of experts reached at this stage is 10 so the sample size was adequate. For the quantitative phase, it was suggested that small sample sizes of 65 and 40 respectively are sufficient for the content validation stage [121, 122]. Therefore, the number of 120 and 400 participants (Sample 1 and Sample 2) was sufficient for the sample of the content validation stage. In a similar vein, the sample calculation formula of Bartlett, Kortlik and Higgins [123]was used for factor construct verification and construct validity stage. Hereunder, the number of remote workers in Türkiye in 2020 was 817,980 [7]. According to the relevant formula, the minimum number of samples required was 384. At this stage, 527 remote workers constituted the sample of the research so the sample size is sufficient.

The inclusion and exclusion criteria were determined by having remote working experience. Accordingly, employees with remote working experience were included. Employees without remote working experience were excluded. In addition, there were two control questions in the questionnaires as "I can not use computer" and "I can not read and write". The participants with wrong answers to at least one control question were excluded, as well. The data was sent to the participants online via Google Forms. Participants answered survey questions via their electronic devices (computer, phone, tablet, etc.) between April 2021—February 2022. All the survey questions were obligatory, therefore there is no missing value in the dataset. Self-report paper-based valid questionnaires were obtained from 1,047 adult remote workers from Türkiye. Participation in this study was voluntary, and nonparticipation did not lead to any disadvantage. The participants were informed orally that their personal information

would be treated anonymously and would remain confidential. In accordance with the Declaration of Helsinki (amended in Fortaleza in 2013), this study was conducted under the approval of the Ethics Committee of Atatürk University (study approval no.: E-88656144-050.01.04–2100142559).

This study was conducted under the guidance of Hinkin [124]. Accordingly, the steps of the scale development and related analyses performed in this study are as follows:

- Step 1 –Item generation: Literature review regarding remote working through a deduction approach, Frequency/Percentage Analyses of the experts' opinions, evaluation of the experts' recommendations.

- Step 2 –Questionnaire administration: Frequency/Percentage Analyses with Sample 1 and Sample 2 for content validation.

- Step 3 –Initial item reduction: Exploratory Factor Analysis (EFA) and parallel analysis with Sample 3.

- Step 4 –Construct verification: Confirmatory Factor Analysis (CFA) with Sample 4.

- Step 5 –Convergent / Discriminant validity: Zero-Order Correlation Analysis of RWSS via Performance and Organizational Commitment Scales with Sample 4. Reliability analysis of the RWSS.

A series of empirical studies for developing and evaluating a five-factor RWSS is given below as; "item generation and content validation (Step 1, Step 2)", "initial factor structure analysis (Step 3)", and "factor construct verification and construct validity analysis (Step 4, Step 5)".

## Item generation and content validation

There are 4 stages in the item generation and content validation process: Study 1, Study 2, Study 3, and Study 4. Study 1 reviews the literature regarding remote working through a deduction approach suggested by Hinkin [124] and the determination of sub-dimensions of RWSS. Therefore, three research questions (RQ1, RQ2, RQ3) were developed regarding remote working skills, and the keywords for each research question were determined and grouped. Keywords of RQ1: "1st Group: Company, business, corporation, organization, 2nd Group: Remote work, telework, online work, e-work, virtual work, 3rd Group: Success, performance, efficiency. Keywords of RQ2: "1st Group: Employee, worker, 2nd Group: Remote work, telework, online work, e-work, virtual work, 3rd Group: Success, performance, efficiency. Keywords of the RQ3: "1st Group: Employee, worker, 2nd Group: Remote work, telework, online work, e-work, virtual work, 3rd Group: Skills. A literature review was conducted with multiple combinations of these three groups of keywords for each research question (An example combination of RQ1 as company, remote work, success).

The relevant search was carried out in Google Scholar and Web of Science. In the studies examined, information related to skills were included and those that were not relevant were excluded. For instance, since the behavioral and personal characteristics required for employees to be successful in remote work [26, 125, 126] are not considered within the scope of skills (e.g. diligence, sociability, need for achievement, need for autonomy, trust, etc.) are excluded. The skills obtained as a result of the review were analyzed and categorized. For example, the skills related to work-life balance are considered as time management skills. Communication skills are divided into verbal and written. As a result, five fundamental skill dimensions were determined. The sub-dimensions of RWSS were determined as security, problem-solving,

time management, verbal communication, and written communication skills, and the initial item pool (49 items) was generated. Study 2 involves asking the opinions of 10 experts (five academics, two senior virtual work managers, and three employees with at least three years of remote working experience) regarding the initial item pool for content validation. An expert's opinion form with three options -"keep," "remove," or "revise as follows"- for each item is appropriate for this purpose. Three additional open-ended blank spaces were included for the experts to provide any additional suggestions. In this study, the acceptable agreement threshold for the experts was established as 80%. According to the opinions of the experts and the author's evaluations regarding these suggested items, 19 items remained, 25 items were revised, 5 items were removed, and 9 new items were added. Therefore, the number of items increased from 49 to 52, and the second item pool was generated. Study 3 presents the second item pool to 120 participants with remote working experience (Sample 1) and the generation of the third item pool. The majority of sample 1's (n = 120) participants were female (60%), between the age range of 22–41 (95%), had 1–3 years of remote working experience (56.7%), and work in the education, health, sports, or other social work sectors (47.5%). The participants were asked to rate the items regarding remote working in order of importance. Data collected through a 7-point Likert-type scale (1 = Not important, 7 = Very important) is more likely of individuals reflect their objective reality because there are more options available [127]. The data collected with the scale was analyzed, and firstly, the name of the "security" dimension was changed to "cybersecurity" because the statements were more related to cybersecurity rather than general security.

As a result, two cybersecurity items were removed (one was removed because the acceptable agreement level was below 80%), and another cybersecurity expression was revised for clarity. In the problem-solving dimension, 2 items were removed because they were below the acceptable agreement level, and another item was removed to prevent repetition. The time management dimension remained the same. The authors evaluated the verbal communication dimension according to the feedback, and 6 items were removed in order to avoid repetition and give clarity. Finally, an item was removed from the written communication dimension to avoid repetition. Therefore, 41 expressions remained for the third item pool.

The purpose of Study 4 was to finish the item generation process and to verify content validation. For this purpose, the third item pool was presented to 400 participants with prior remote working experience (Sample 2). The majority of sample 2 (n = 400) participants were male (54.5%), between the age range of 22–41 (76%), have 1–3 years of remote working experience (66.5%), not a manager (68.5%), and work in the technology sector (27%). They were asked to rate the items in order of importance through a 7-point Likert type (1 = Not important, 7 = Very important) scale. At the end of the analysis of the obtained findings, three items (2 from the verbal communication dimension and 1 from the written communication dimension) were removed because they were below the acceptable agreement level. Therefore, the number of items decreased from 41 to 38.

## RWSS initial factor structure analysis

875 participants were contacted for the common purposes of Study 5 and Study 6. At this stage, support from a professional research company was gathered in the data collection process. 233 participants were eliminated because they did not have remote working experience; 115 participants were eliminated because they failed to pass at least 1 out of 2 attention control tests ("I do not use computers" and "I am illiterate"). The remaining 527 participants were randomly divided into two sample groups. Sample 3 (n = 263) was used for Study 5, and Sample 4 (n = 264) was used for Study 6. According to Hinkin [124], the split-half method is an

acceptable method for both EFA and CFA. The majority of participants in Sample 3 (n = 263) are female (66.9%), between the age range of 26 and 35 (53.6%), and have 1–2 years of remote working experience (68.8%); they are not managers (78.7%), and they work in education, healthcare, sports, and other social sectors (45.2%). The majority of participants in Sample 4 (n = 264) are female (69.3%), between the age range of 26 and 35 (52.7%), and have 1 year or less remote working experience (93.2%); they are not managers (89%), and they work in education, healthcare, sports, and other social sectors (34.8%). Study 5 is a part of the RWSS initial factor construct analysis stage. Explanatory Factor Analysis (EFA) was performed at this stage with Sample 3. EFA is a crucial step for scale development studies to provide evidence for construct verification and reduce the number of items [124]. EFA scores obtained on the RWS scale were analyzed using the principal axis method with oblique factor rotation (Promax), which should be used for evaluating non-normal data. The following criteria were used to validate the new scale: theoretical considerations, the Kaiser-Meyer-Olkin and Bartlett's test of sphericity, scree plot, eigenvalues, and the percentage of variance explained.

In addition to the K1 method [128], Parallel Analysis, which is another analysis method developed by Horn [129], was utilized to support the five-factor structure of the scale. SPSS Statistics 22 program was used for parallel analysis and SPSS Syntax developed by O'Connor [130] was used to perform the analysis. It can be determined the ideal number of factors by using parallel analysis [131]. The analysis compares the eigenvalues of the real data set and the random data set generated in parallel, and the number of significant factors is accepted as the last eigenvalue point of the real data set that is greater than the random data set [129].

## RWSS factor construct verification and construct validity analysis

Within the parameters of Study 6, RWSS factor construct verification and construct validity analysis were performed. At this stage, Confirmatory Factor Analysis (CFA) was performed with Sample 4 to test the overall suitability of the construct. In order to examine the factor structure of the new measure, CFA was conducted. Each item was therefore input as an observed variable and loaded onto its corresponding latent factor (cybersecurity skills, problem-solving skills, time management skills, verbal communication skills, and written communication skills). Subsequently, the convergent and discriminant validity was evaluated to provide evidence of the construct validity of the RWSS. Zero-order correlation analysis was performed with RWSS via Performance and Organizational Commitment Scales for the convergent and discriminant validity [132].

A four-item version ($\alpha$ = .93) of the Employee Performance Scale [133] was used. "I am exceeding my goals" is a sample statement. For Affective and Continuance Commitment, the Affective and Continuance sub-scales of the Organizational Commitment Scale, developed by Meyer and Allen [112] and adapted into Turkish by Baysal and Paksoy [134], were used. The Affective Commitment Scale includes six items ($\alpha$ = .95), such as "I feel like a part of the family at my organization", and the Continuance Commitment Scale consists of six items ($\alpha$ = .85), such as "continuing to work at my current organization is both my desire and obligation". The participants gave their opinions using a 7-point Likert-type scale ranging between 1 (Strongly Disagree) and 7 (Strongly Agree). Higher scores denote higher performance and higher levels of affective and continuance commitment.

For the convergent validity evidence, it was specifically anticipated that the RWSS would demonstrate medium and high positive correlations with performance and small and medium positive correlations with affective commitment. For the discriminant validity, age, gender, and continuance commitment are expected to be uncorrelated with RWSS. The psychometric properties of the Performance and Commitment scales used in discriminant and

convergent validity were analyzed to test their suitability for correlation analysis. In addition to Cronbach's Alpha, Composite reliability, Average Variance Extracted (AVE) values, and standardized factor loadings [135, 136] were all utilized for the psychometric properties of the organizational commitment and performance scales. Finally, in order to provide the reliability of RWSS, composite reliability was examined in addition to Cronbach's Alpha. Composite reliability (CR) formula [135] and standardized factor loadings of RWSS were used to provide CR.

## Results

### Exploratory factor analysis

EFA with 38 items resulted in a five-factor model. Following EFA, two items were removed due to the cross-loadings. Final EFA was performed with the remaining 36 items, resulting in the hypothesized five-factor structure, as given in Table 1 below. Factor loadings ranging from.380 to.982 explained 68.347% of the total variance. Tabachnick and Fidell [137] suggested that the minimum factor loading should be.30. Therefore the factor loadings are at sufficient level. Each factor has high internal consistency reliabilities ($\alpha > 0.70$, [138]), with 5 items for cybersecurity skills ($\alpha = .883$), 8 items for problem-solving skills ($\alpha = .950$), 5 items for time management skills ($\alpha = .892$), 7 items for verbal communication skills ($\alpha = .903$), and 11 items for written communication skills ($\alpha = .956$).

Parallel Analysis results are displayed in Table 2. As Table 2 is examined, the last point where the eigenvalue of the real data set (0.833) is greater than the eigenvalue of the random data set (0.677) is 5. According to Horn [129], this result demonstrated that the scale has five factors. Thus, the scale's five-factor structure was verified.

### Confirmatory factor analysis

According to CFA results, factor loadings ranged from 0.63 to 0.92 (See Table 3). The obtained model fit indices are $\chi2 = 1384$, $\chi2/df = 2.391$, CFI = 0.91, TLI = 0.90, IFI = 0.91, SRMR = 0.056, RMSEA = 0.073(%90 confidence interval, lower bound = 0.068, upper bound = 0.078). Model fit indices obtained indicated a fit ($\chi2/df < 3$, CFI > 0.90, TLI > 0,90, IFI > 0.90, 0.05 < SRMR < 0.08, 0.05 < RMSEA < 0.08) with the data [139–142]. According to CFA, RWSS provided a five-factor model structure as shown in Fig 1.

### Psychometric properties of organizational commitment and performance scales

The formula in Fig 2 was used to calculate the AVE values using the standardized factor loadings of the scale (See Table 4).

### Zero-order correlation analysis

Once investigating the Spearman's correlation matrix for the variables in Sample 4 (see Table 5), the subscales cybersecurity skills, r = 0.41, p < .01, problem-solving skills, r = 0.48, p < .01, time management skills, r = 0.73, p < .01, verbal communication skills r = 0.48, p < .01, written communication skills, r = 0.47, p < .01 and the overall RWSS r = 0.59, p < .01 were all positively related to performance. Following this, the subscales of cybersecurity skills, r = 0.30, p < .01, problem-solving skills, r = 0.32, p < .01, time management skills, r = 0.45, p < .01, verbal communication skills r = 0.24, p < .01, written communication skills, r = 0.27, p < .01, and the overall RWSS r = 0.37, p < .01 were all positively related to affective commitment. According to the results of the correlation between age, gender, continuance commitment and RWSS,

**Table 1. Exploratory factor analysis results.**

| Item | M | SD | Cybersecurity skills | Problem solving skills | Time management skills | Verbal communication skills | Written communication skills |
|---|---|---|---|---|---|---|---|
| I can protect digital decives physically. | 6.09 | 1.21 | 0.38 | | | | |
| I can implement the security strategy recommended by the institution. | 6.25 | 1.04 | 0.63 | | | | |
| I can provide digital privacy of mine and others. | 5.97 | 1.31 | 0.86 | | | | |
| I can protect the digital device from external threats such as viruses etc. | 5.46 | 1.52 | 0.66 | | | | |
| I can provide data privacy. | 5.81 | 1.42 | 0.73 | | | | |
| | | | | | | | |
| I can choose the appropriate tool and method to solve problems (technical and non-technical). | 5.89 | 1.21 | | 0.57 | | | |
| I can solve problems that arise when the current technologies do not work. | 5.63 | 1.33 | | 0.60 | | | |
| I can select appropriate technology (tool, device, application, software, service, etc.) required by the task. | 5.90 | 1.31 | | 0.68 | | | |
| I can choose a tool that fits the purpose and evaluate the tool's effectiveness. | 6.04 | 1.15 | | 0.59 | | | |
| I can use new technological devices (new software, interfaces, hardware, etc.). | 5.83 | 1.32 | | 0.77 | | | |
| I can learn to do something new with current technologies. | 6.11 | 1.17 | | 0.73 | | | |
| I can keep digital competencies up to date. | 5.97 | 1.29 | | 0.84 | | | |
| I can manage efficiently the excessive flow of information brought by information and communication technologies. | 5.87 | 1.21 | | 0.77 | | | |
| I can calculate accurately how long the tasks will take. | 6.23 | 1.06 | | | 0.62 | | |
| I can sort tasks by importance and degree of urgency. | 6.43 | 0.90 | | | 0.63 | | |
| I can use the time worked online (working hours defined by the employer) effectively. | 6.35 | 1.00 | | | 0.79 | | |
| I can maintain work-life balance. | 5.92 | 1.24 | | | 0.81 | | |
| I can cope with distractions at home (demands of family members, needs of pets, housework, noise, etc.). | 5.92 | 1.32 | | | 0.72 | | |
| I can participate in the online conversations and dialogues. | 6.15 | 1.30 | | | | 0.62 | |
| I can resolve the conflicts in online environments. | 5.86 | 1.25 | | | | 0.74 | |
| I can receive feedback. | 6.04 | 1.22 | | | | 0.73 | |
| I can give feedback. | 6.19 | 1.13 | | | | 0.82 | |
| I can communicate correctly in times of crises. | 5.98 | 1.25 | | | | 0.81 | |
| I can focus on the main points (avoiding unnecessary details, keeping words, being simple, etc.). | 6.25 | 1.08 | | | | 0.55 | |
| I can listen actively (asking questions, trying to understand, focusing, etc.). | 6.34 | 1.04 | | | | 0.41 | |
| I can spell the words correctly. | 6.60 | 0.90 | | | | | 0.91 |
| I can use the grammar correctly. | 6.53 | 0.92 | | | | | 0.98 |
| I can use a simple, easy, understandable and fluent language. | 6.52 | 0.90 | | | | | 0.96 |
| I can express ideas clearly. | 6.50 | 0.86 | | | | | 0.69 |

*(Continued)*

**Table 1.** (*Continued*)

| Item | M | SD | Cybersecurity skills | Problem solving skills | Time management skills | Verbal communication skills | Written communication skills |
|------|---|----|----|----|----|----|----|
| I can write in a way that the reader can understand. | 6.59 | 0.82 | | | | | 0.82 |
| I can transmit information accurately (to the right person, at the right time, with the right tool, etc.). | 6.56 | 0.82 | | | | | 0.83 |
| I can write in an appropriate format for different readers (employees, customers, suppliers, public institutions, etc.). | 6.28 | 1.05 | | | | | 0.58 |
| I can use verified information from different resources to ensure the accuracy of the content. | 6.44 | 0.95 | | | | | 0.60 |
| I can use a professional writing style. | 6.17 | 1.16 | | | | | 0.70 |
| I can write clear instructions. | 5.97 | 1.31 | | | | | 0.54 |
| I can express myself quickly and accurately while using instant messaging tools (Whatsapp, Telegram, etc.). | 6.50 | 0.93 | | | | | 0.59 |
| % of variance (rotated solution) | | | 12.4 | 14.5 | 12.4 | 13.9 | 15.7 |
| Alpha coefficient | | | 0.88 | 0.95 | 0.89 | 0.90 | 0.96 |

Note: N = 263. Interfactor correlations range from.54 to.70. Principal axis factor analysis with Promax rotation.

**Table 2.** Parallel analysis eigenvalues results.

| No | Real Data Set Eigenvalues | Mean | Random Data Set Eigenvalues %95 |
|----|----|----|----|
| 1 | 19,18833 | 0,922036 | 1,02755 |
| 2 | 2,320549 | 0,824736 | 0,90552 |
| 3 | 1,420487 | 0,747044 | 0,815802 |
| 4 | 1,067065 | 0,681143 | 0,740871 |
| 5 | 0,83302 | 0,623118 | 0,67726 |
| 6 | 0,551314 | 0,570327 | 0,621569 |
| 7 | 0,4857 | 0,521192 | 0,57428 |
| 8 | 0,338855 | 0,474098 | 0,52382 |
| 9 | 0,325462 | 0,431304 | 0,477837 |
| 10 | 0,288915 | 0,389545 | 0,433685 |
| 11 | 0,201964 | 0,348343 | 0,391182 |
| 12 | 0,154543 | 0,309262 | 0,349179 |
| 13 | 0,137297 | 0,271269 | 0,309715 |
| 14 | 0,109502 | 0,235932 | 0,274404 |
| 15 | 0,095427 | 0,200984 | 0,238993 |
| 16 | 0,072839 | 0,167281 | 0,2028 |
| 17 | 0,054825 | 0,133436 | 0,166792 |
| 18 | 0,046454 | 0,100567 | 0,134201 |
| 19 | 0,039218 | 0,069362 | 0,100864 |
| 20 | 0,007875 | 0,03757 | 0,067952 |

**Table 3. Confirmatory factor analysis results.**

| Item | *M* | *SD* | Cybersecurity skills | Problem solving skills | Time management skills | Verbal communication skills | Written communication skills |
|------|-----|------|----------------------|------------------------|------------------------|-----------------------------|------------------------------|
| **CSS1** | 6.01 | 1.30 | 0.740 | | | | |
| **CSS2** | 6.12 | 1.19 | 0.877 | | | | |
| **CSS3** | 6.09 | 1.22 | 0.867 | | | | |
| **CSS4** | 5.30 | 1.57 | 0.638 | | | | |
| **CSS5** | 5.77 | 1.51 | 0.726 | | | | |
| **PSS1** | 5.73 | 1.41 | | 0.756 | | | |
| **PSS2** | 5.41 | 1.49 | | 0.732 | | | |
| **PSS3** | 5.75 | 1.37 | | 0.830 | | | |
| **PSS4** | 5.86 | 1.25 | | 0.844 | | | |
| **PSS5** | 5.61 | 1.44 | | 0.825 | | | |
| **PSS6** | 5.93 | 1.31 | | 0.871 | | | |
| **PSS7** | 5.92 | 1.27 | | 0.880 | | | |
| **PSS8** | 5.74 | 1.29 | | 0.903 | | | |
| **TMS2** | 6.18 | 1.13 | | | 0.896 | | |
| **TMS3** | 6.38 | 1.06 | | | 0.908 | | |
| **TMS4** | 6.31 | 1.12 | | | 0.907 | | |
| **TMS5** | 5.92 | 1.41 | | | 0.746 | | |
| **TMS6** | 5.84 | 1.43 | | | 0.650 | | |
| **VCS1** | 5.78 | 1.58 | | | | 0.676 | |
| **VCS3** | 5.50 | 1.41 | | | | 0.787 | |
| **VCS5** | 5.90 | 1.22 | | | | 0.820 | |
| **VCS6** | 6.02 | 1.27 | | | | 0.806 | |
| **VCS7** | 5.87 | 1.23 | | | | 0.758 | |
| **VCS8** | 6.15 | 1.11 | | | | 0.774 | |
| **VCS9** | 6.29 | 1.16 | | | | 0.693 | |
| **WCS1** | 6.53 | 1.01 | | | | | 0.836 |
| **WCS3** | 6.42 | 1.05 | | | | | 0.862 |
| **WCS4** | 6.47 | 1.00 | | | | | 0.924 |
| **WCS5** | 6.51 | 1.03 | | | | | 0.853 |
| **WCS6** | 6.53 | 0.97 | | | | | 0.917 |
| **WCS7** | 6.51 | 0.96 | | | | | 0.893 |
| **WCS8** | 6.20 | 1.22 | | | | | 0.801 |
| **WCS9** | 6.40 | 1.08 | | | | | 0.802 |
| **WCS10** | 6.03 | 1.28 | | | | | 0.771 |
| **WCS11** | 5.92 | 1.26 | | | | | 0.699 |
| **WCS12** | 6.30 | 1.13 | | | | | 0.737 |

Note: N = 264. Based on the Maximum Likelihood. Inter-factor correlations range from 0.64 to 0.79. Model fit indices: $\chi^2$ = 1384, $\chi^2/df$ = 2.391, CFI = 0.91, TLI = 0.90, IFI = 0.91, SRMR = 0.056, RMSEA = 0.073.

Abbreviations: CSS: Cybersecurity Skills, PSS: Problem-Solving Skills, TMS: Time Management Skills, VCS: Verbal Communication Skills, WCS: Written Communication Skills.

age, gender, continuance commitment did not correlate with the dimension of the RWSS (except for small correlations with gender and written communication skills, r = 0.18, p < .01) or with the overall scale. Considering all these assumptions (H2, H3, H4) and results, RWSS provided convergent and discriminant validity.

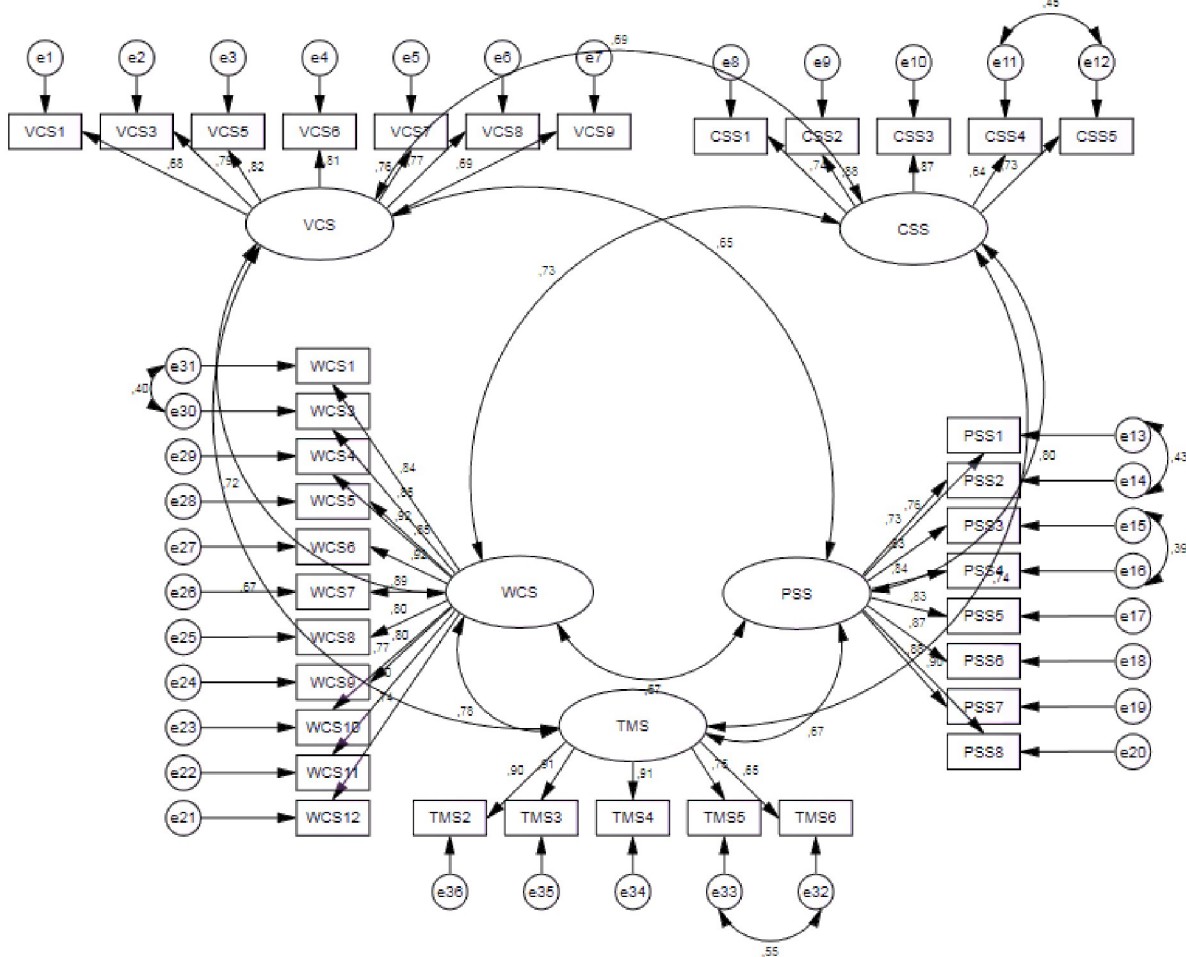

**Fig 1. CFA model.**

$$\rho_{vc(\eta)} = \frac{\sum_{i=1}^{\rho} \lambda_{yi}^2}{\sum_{i=1}^{\rho} \lambda_{yi}^2 + \sum_{i=1}^{\rho} Var(\varepsilon_i)}$$

**Fig 2. AVE formula.** To ensure convergent validity, AVE values above 0,50 are sufficient [136]. Values must be greater than 0.6 to ensure composite reliability [135]. AVE values of the scales (Continuance commitment scale = 0,5004; Affective commitment scale = 0,79; Performance scale = 0,77) and Composite reliability values (Continuance commitment = 0,85; Affective commitment = 0,95; Performance = 0,93) were both above 0,6. These results showed that the performance and commitment scales were appropriate for analysis.

**Table 4. Standardized factor loadings of commitment scales and performance scale.**

| | Continuance commitment | Affective commitment | Performance |
|---|---|---|---|
| **Item** | Standardized Factor Loadings | | |
| **1** | 0,754 | 0,891 | 0,841 |
| **2** | 0,746 | 0,948 | 0,896 |
| **3** | 0,680 | 0,954 | 0,904 |
| **4** | 0,763 | 0,934 | 0,875 |
| **5** | 0,668 | 0,839 | |
| **6** | 0,622 | 0,782 | |

## Reliability analysis

The formula in Fig 3 [135] was applied by utilizing the standardized factor loadings of the RWSS (See Table 3) to calculate the composite reliability values. According to the results of the formula applied in line with standardized factor loadings, all factors that make up the RWSS had composite reliability values that were higher than 0.6 (VCS = 0.90; CSS = 0.88; PSS = 0.94; WCS = 0.96; TMS = 0.91). Values must be greater than 0.6 to ensure composite reliability [135]. All factors that make up the RWSS had composite reliability values that were higher than 0.6 (VCS = 0.90; CSS = 0.88; PSS = 0.94; WCS = 0.96; TMS = 0.91). Accordingly, the composite reliability of the scale was provided.

## Discussion

Scale development stages were carried out according to the guidance of Hinkin [124] and the measurement criteria of each stage were fulfilled. Thus, the validity and reliability of the RWSS were provided. As a result of factor analysis and parallel analysis, H1 was accepted and the five-factor structure of the scale was proven. As a result of the correlation analysis for

**Table 5. Zero-order correlations among study variables.**

| Variables | M | SD | 1 | 2 | 3 | 4 | 5 | 6 | 7 | 8 | 9 | 10 | 11 |
|---|---|---|---|---|---|---|---|---|---|---|---|---|---|
| **1.Gender** | %69 | Female | | | | | | | | | | | |
| **2.Age** | 1.86 | 0.71 | 0.18** | | | | | | | | | | |
| **3.Cybersecurity** | 5.85 | 1.12 | 0.02 | -0.12 | **0.88** | | | | | | | | |
| **4.Problem solving** | 5.74 | 1.16 | 0.04 | -0.09 | 0.67** | **0.94** | | | | | | | |
| **5.Time management** | 6.12 | 1.06 | -0.07 | 0.02 | 0.49** | 0.54** | **0.91** | | | | | | |
| **6.Verbal communication** | 5.92 | 1.02 | 0.03 | 0.08 | 0.50** | 0.56** | 0.49** | **0.90** | | | | | |
| **7.Written communication** | 6.34 | 0.91 | -0.14* | -0.07 | 0.49** | 0.50** | 0.51** | 0.54** | **0.95** | | | | |
| **8.Performance** | 6.24 | 0.97 | -0.06 | 0.06 | 0.41** | 0.48** | 0.73** | 0.49** | 0.48** | **0.93** | | | |
| **9.Affective commitment** | 5.50 | 1.54 | -0.01 | 0.15* | 0.30** | 0.33** | 0.46** | 0.25** | 0.28** | 0.43** | **0.95** | | |
| **10.Continuance commitment** | 4.39 | 1.54 | 0.01 | -0.01 | 0.06 | 0.01 | -0.04 | -0.02 | -0.04 | -0.01 | 0.09 | **0.85** | |
| **11.RWSS (Overall)** | 6.03 | 0.89 | -0.01 | -0.05 | 0.78** | 0.87** | 0.69** | 0.78** | 0.75** | 0.60** | 0.37** | -0.04 | **0.97** |

Note: N = 264,

*denotes p < .05,

** indicates p < .01.

Gender coded 1 = Female, 2 = Male. Age Coded 1 = 25 and under, 2 = 26–35, 3 = 36–45, 4 = 46 and above. Bold numbers on the diagonal indicate internal consistency (Cronbach's Alpha)

$$\rho_c = \frac{(\sum \lambda_i)^2 \, var(T)}{(\sum \lambda_i)^2 \, var(T) + \sum \theta_{ii}}$$

**Fig 3. Composite reliability formula.**

discriminant and convergent validity with RWSS via Performance and Organizational Commitment Scales, H2, H3, and H4 were accepted.

In line with H2, it has been proven that remote working skills are positively related to performance and therefore make remote working successful. This result supports previous studies in the literature [16, 23, 111]. Likewise, in line with H3, it has been proven that remote working skills are positively related to affective commitment and thus make remote working successful. This result supports previous studies in the literature [113–116].

In line with H4, it has been proven that remote working skills do not have a significant relationship with continuance commitment. This result supports previous studies in the literature [117, 118]. At the same time, the results obtained show that it is compatible with Resource-based Theory [29] and Human Capital Theory [30], which considers skills as a valuable resource in the success of organizations. In this context, it has been demonstrated that the developed RWSS is an important tool in the success of the remote working model.

## Theoretical and practical implications

Today, a growing number of organizations adopt completely remote or hybrid working models. An increasing number of virtual organizations have been working remotely. There are no significant differences between working remotely for a physical office or a virtual organization. In both situations, the working manners of the employees, rather than who they are, will be the point of focus [143], and they will be expected to possess certain skills and competencies for an efficient remote working model. Therefore, employees and managers will both have to develop new skills and capabilities in order to accommodate these new work models [144].

Due to the rise of the information economy, there is an increased number of remote and virtual working models, which has led to increased competition among employees and candidates with the required skills and competencies. Employers are frequently willing to employ individuals with the needed skills and offer them inviting rewards, such as higher salaries and promotion opportunities. However, there is still a skill deficit among employers and job seekers. Skill deficit has long been considered as the main reason behind inequality in many job postings and underemployment worldwide. It is argued that accurately determining the skills in human capital data is the first step in resolving this socioeconomic problem [145]. In the study conducted by Braesemann et al. [18], it was revealed that remote working skills are not evenly distributed around the world, there is a polarization, and southern countries lag behind northern countries and rural areas lag behind cities. In addition, it has been found that employees who do not have sufficient skills in remote work receive less salary than employees who have these skills and are not in equal competitive conditions. In light of these findings, it is thought that the remote working skills and scale revealed in our study can be used as a tool to equalize competition in remote work, increase employment, and develop the workforce, and thus contribute to the economic development of countries.

In this study, remote working skills were identified, along with the skills required for IT employees, skills for the 21st century, etc., (e.g., [14, 22, 51, 56, 70, 87–89, 92, 95, 145–147])

were analyzed, remote working skills were determined, and a five-dimensional scale was developed. It is believed that a scale, which will be used to determine and measure remote working skills, will be very useful and fill a significant gap in remote working literature. Due to the increasing remote working activities, human resources management offices are faced with the need for the requalification of employees [148]. The executives can assess their employees' cybersecurity, problem-solving, time management, verbal communication, and written communication skills using RWSS and train and develop their personnel accordingly. Through the RWSS, employees can identify the skills they need to achieve their career goals, especially in this working model, and thus use the RWSS as a guide to obtain and develop these skills. The skills required for remote working, which is today's reality and the new normal of the future, should be taught to the new generations. As a result, educational institutions at any level can benefit from the outcomes of this study.

## Limitations and future research directions

In this study, a considerable part of the articles within significant databases regarding remote working were reviewed, and the authors could not find any study that directly focuses on remote working skills. The authors argue that the remote working skills represent a valid perspective, and that the developed scale will measure these skills pertinently and reliably. These skills were developed based on pertinent studies in the literature that discuss and mention remote working skills. Participants in this study were assumed to have digital and professional expertise. These were not included in the remote working skills because these employees would not be able to work from the office without professional expertise, which would not be considered a distinctive factor in remote working. Additionally, it is impossible to work remotely without the use of computers and other related technologies. It was also determined that the employees' traits would not make any contributions within this context, and the authors preferred to concentrate on improvable skills.

The population of this research consists of Turkish-speaking participants from Türkiye. It is of course possible to consider this as a constraint. However, it should be noted that 1.047 individuals involved in four different samples within this study work for national, international, and global companies that offer remote working opportunities. Considering the generalizability of the scale, Türkiye's place in global trade and working conditions are important in this regard. Türkiye has an important share in international trade. As well as Turkish companies operating globally, the number of international and foreign capital companies operating in Türkiye is 76,737 as of the end of 2021 [149]. A significant part of the participants in our study consisted of people working in these companies. In light of this information, it can be stated that Türkiye's working conditions are at global standards and therefore the skills required in both face-to-face work and remote work are global. In this respect, although the remote working skills scale was developed in Turkish, it is thought that the English version of the scale can be applied globally. On the other hand, for researchers who wish to use the scale in their national language, the RWSS can be adapted by considering the cultural elements.

This study presents the perspective of those who have personally experienced (self-reported surveys) remote work. Psychologists also primarily assess non-cognitive skills by using self-reported surveys or observer reports [150]. The emphasis in this direction is on identifying the requirements that employees need to possess in addition to the other requirements that will ensure person-job fit when they have to work remotely. Therefore, the study should be reviewed from this perspective. RWSS includes remote working skills, which are considered to be required within the context of today's practices. All indicators show that the business world

will mostly opt for a hybrid working model, which involves working both remotely and from the office.

As the number of organizations adopting the remote working model increases, it will become more important to understand the most useful skills for various groups of people under different working conditions to build up-and-running work environments, and more data will be required. Along with being informed about remote working skills with contributions of RWSS, it will be crucial to know how to acquire these skills at various educational levels. Therefore, it may be recommended that future studies work on acquiring these remote working skills. Last but not least, the relationships and interactions between the skills revealed in this study and other important factors such as individual (personal traits, attitudes), the job (the nature of the job, technology, etc.), the organization (strategy, culture, etc.), and home-family [12] which are important in the remote working model, can be investigated. These future studies can contribute both theoretically and practically to the development of the remote working model.

## Conclusion

The purpose of this study is to determine the skills required for efficient remote working activities and to develop a scale regarding these skills. To achieve this, the items regarding remote working skills were generated, and content validation was performed, afterwards, item reduction was applied, the five-factor structure of the scale was verified, and finally, convergent and discriminant validations were made. Finally, the RWSS, consisting of 36 items and five dimensions, was developed. These dimensions are cybersecurity (five items), problem-solving (eight items), time management (five items), verbal communication (seven items), and written communication (eleven items).

## Supporting information

**S1 File.**
(DOCX)

**S1 Data.**
(ZIP)

## Acknowledgments

We dedicate this manuscript to remote employees around the world. We are grateful to the experts who have supported our scale development study through their experiences and ideas. Lastly, we are grateful to the participants who took the time to complete our questionnaires.

## Author Contributions

**Conceptualization:** Serap Benligiray, Abdullah Y. Güngör, İlkay Akbaş.

**Data curation:** Serap Benligiray, Abdullah Y. Güngör, İlkay Akbaş.

**Formal analysis:** Serap Benligiray, Abdullah Y. Güngör, İlkay Akbaş.

**Methodology:** Serap Benligiray, Abdullah Y. Güngör, İlkay Akbaş.

**Supervision:** Serap Benligiray.

**Validation:** Serap Benligiray, Abdullah Y. Güngör, İlkay Akbaş.

**Writing – original draft:** Serap Benligiray, Abdullah Y. Güngör, İlkay Akbaş.

**Writing – review & editing:** Serap Benligiray, Abdullah Y. Güngör, İlkay Akbaş.

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
