## [Decision Letter · Decision Letter 0]

30 Oct 2023

PONE-D-23-23682Measuring remote working skills: Scale development and validation studyPLOS ONE

Dear Dr. GÜNGÖR,

Thank you for submitting your manuscript to PLOS ONE. After careful consideration, we feel that it has merit but does not fully meet PLOS ONE’s publication criteria as it currently stands. Therefore, we invite you to submit a revised version of the manuscript that addresses the points raised during the review process.

We look forward to receiving your revised manuscript.

Kind regards,

Dan-Cristian Dabija, PhD

Academic Editor

PLOS ONE

4. Please ensure that you include a title page within your main document. You should list all authors and all affiliations as per our author instructions and clearly indicate the corresponding author.

5. We notice that your supplementary figures are uploaded with the file type 'Figure'. Please amend the file type to 'Supporting Information'. Please ensure that each Supporting Information file has a legend listed in the manuscript after the references list.

6. We notice that your supplementary figures and tables are included in the manuscript file. Please remove them and upload them with the file type 'Supporting Information'. Please ensure that each Supporting Information file has a legend listed in the manuscript after the references list.

Additional Editor Comments:

Dear authors

the reviewers consider that the manuscript has merits, but that some changes are needed. Please implement all their comments.

Thanks

Reviewers' comments:

Reviewer's Responses to Questions

**Comments to the Author**

1. Is the manuscript technically sound, and do the data support the conclusions?

Reviewer #1: Yes

Reviewer #2: Partly

2. Has the statistical analysis been performed appropriately and rigorously? 

Reviewer #1: Yes

Reviewer #2: Yes

3. Have the authors made all data underlying the findings in their manuscript fully available?

Reviewer #1: Yes

Reviewer #2: No

4. Is the manuscript presented in an intelligible fashion and written in standard English?

Reviewer #1: Yes

Reviewer #2: Yes

5. Review Comments to the Author

Reviewer #1: I have read the referred article with keen interest. The information is interesting and innovative; conclusion section is interesting and authors can improve it further. I am recommending authors to do a little more work and add latest literate to support the study. The authors need to improve results section. The level of English is good and smooth, e.g., the language standard, specifically the grammar, of sufficient quality to meet scientific merit for publication. However, I suggest authors to double check for language quality. Describe scientific contribution of the study to the existing body of knowledge. I endorse this manuscript after minor revision as suggested. The topic is interesting and worthy of attention. The methodology is adequate and the conclusions are consistent with the reported data. The manuscript can be improved by expanding the references and citing some recently published articles on this topic.

Reviewer #2: This study surveyed remote workers in Turkey and developed a scale that allows workers to self-assess their remote working skills. This scale is important for maintaining and improving workers' performance and preventing various negative physical, mental, and family effects. This is because the need to work remotely is increasing worldwide due to the promotion of ICT in labor and the expansion of remote work in the wake of the pandemic of the new coronavirus infection.

Although this manuscript is interesting and significant, there are some points lacking in the overall description, such as the introduction, methods, results, and discussion. In particular, the methods, results, and discussion do not follow the COSMIN Reporting Guideline for studies on measurement properties.

Specific comments are as follows；

1． The definition of the concept measured by the scale is ambiguous. What is the author's definition of REMOTE WORKING? What is the definition of the concept that the development scale measures? Please clearly explain these within the Introduction or Methods section of the text.

2． Introduction: The Introduction section is long. It would be better to review and structure the development of the argument.

-While I understand that the authors have conducted a careful literature review, the conceptual structure of the scale and the rationale theory are somewhat unclear, and the explanations of the sub-concepts of the development scale appear to be disjointed.

- Could you please explain a little more about the necessity of developing this scale? For example, in previous studies, is there a scale that can measure remote working skills or not? Does it exist globally or is there no Turkish version? If there is an existing remote working scale, what are the limitations of the existing scale?

3．Methods：Missing are descriptions of the study design, inclusion and exclusion criteria for subjects, data collection methods, sample size calculation methods, and statistical methods for handling missing values.

-I can't find a description of the research design. Is it a cross-sectional design? The type of study being used to test the properties (e.g., test-retest design, longitudinal study, cohort, cross sectional, etc.).

- Please state how the participants were chosen. Please explain the inclusion and exclusion criteria for the study subjects.

-Data collection procedures are missing. An explicit description of how and when the participants were administered (e.g., in what setting) including data collection devices/system used (e.g. paper based, electronic administration) should be provided.

-Please describe sample size calculation.

-The description of statistical methods in the Methods section is too brief. Also, although the statistical methods are described in the Results section, they should be clearly stated in the Methods section.

-In particular, please add to the methods how the validity and reliability of the scale was determined and the statistical values set for the evaluation.

-How did the authors handle missing values?

4．Results: Descriptions of statistical methods in the Results section should be moved to the Methods section.　I think the results should include the CFA Figures.

5．Discussion: The discussion section is too short and does not include a summary of the study, interpretation of the results, the meaning and significance of the results of this study compared to previous studies, the novelty and generalizability of this scale.

-The authors should compare the result to the criteria for good measurement properties (e.g., COSMIN criteria), and determine if the specific Measurement Property is sufficient or not.

-Generalizability issues related to the results should be discussed. For example, discuss if the results could be generalized to other populations given the sample studied.

-Although we have mentioned the development of the Turkish version as a limitation of this project, what are your thoughts on the development of an English version and the possibility of its use in other countries in the world?

-It would be desirable to state in the introduction and discussion whether remote work and workers' remote work skills in Turkey are common to other countries in the world, and whether there are any characteristics or differences among them compared to other countries in the world.

6. PLOS authors have the option to publish the peer review history of their article (what does this mean?). If published, this will include your full peer review and any attached files.

Reviewer #1: **Yes: **Muhammad Aqeel

Reviewer #2: No

---

## [Author Response · Author response to Decision Letter 0]

26 Dec 2023

Response to Reviewers

We are grateful to the editor and reviewers for spending their time to make significant and valuable recommendations for the manuscript. We scrutinized all comments and tried to respond to them step by step as below.

Reviewer #1: I have read the referred article with keen interest. The information is interesting and innovative; conclusion section is interesting and authors can improve it further. I am recommending authors to do a little more work and add latest literate to support the study. The authors need to improve results section. The level of English is good and smooth, e.g., the language standard, specifically the grammar, of sufficient quality to meet scientific merit for publication. However, I suggest authors to double check for language quality. Describe scientific contribution of the study to the existing body of knowledge. I endorse this manuscript after minor revision as suggested. The topic is interesting and worthy of attention. The methodology is adequate and the conclusions are consistent with the reported data. The manuscript can be improved by expanding the references and citing some recently published articles on this topic.

- It was tried to avoid repetition in the conclusion to keep it brief because the conclusion section is optional according to the manuscript organization of PlosONE. However, instead of expanding the conclusion section, the discussion section was expanded in detail, which is closely related to each other.

-The latest studies in the literature were added to the final version of the manuscript as below: Xiong et al., 2023; Zhang et al., 2023; Başol and Çömlekçi, 2022; Braesemann et al., 2022; Tramontano et al., 2021; Grant et al., 2019.

-The result section was revised and extra reference values were added in this section. In addition, the descriptions of statistical methods were moved to the Methods from Results section for clarity.

-Many revisions were implemented after a double language check.

-The necessity of our scale was stated by comparing previous studies in lines 53-72. In addition, the meaning and significance of the results of this study compared to previous studies were explained in the Discussion section, as well, in lines 589-608.

Reviewer #2: This study surveyed remote workers in Turkey and developed a scale that allows workers to self-assess their remote working skills. This scale is important for maintaining and improving workers' performance and preventing various negative physical, mental, and family effects. This is because the need to work remotely is increasing worldwide due to the promotion of ICT in labor and the expansion of remote work in the wake of the pandemic of the new coronavirus infection.

Although this manuscript is interesting and significant, there are some points lacking in the overall description, such as the introduction, methods, results, and discussion. In particular, the methods, results, and discussion do not follow the COSMIN Reporting Guideline for studies on measurement properties.

Specific comments:

1- The definition of the concept measured by the scale is ambiguous. What is the author's definition of REMOTE WORKING? What is the definition of the concept that the development scale measures? Please clearly explain these within the Introduction or Methods section of the text.

-Definitions in the literature were added in the lines of 99-109. Our definition of remote working is in the lines of 109-111. In addition, our definition of remote working skills is in the lines of 140-143. All additions to Response 1 were made in the Introduction section.

2- Introduction: The Introduction section is long. It would be better to review and structure the development of the argument.

-We agree with the comment that the introduction section is long. However, the manuscript organization of PlosONE starts with the introduction section followed by the materials and methods section which are obligatory in order. Therefore, concepts (remote working, remote working skills, cybersecurity skills, problem-solving, time management, verbal communication, and written communication), literature review, and hypotheses are within the introduction section. Except for them, the part we consider as the main introduction of our study is in lines 18-83. 

-However as it was recommended, the development of the argument was reviewed and we tried to improve its structure by adding recent studies, related theories, and research questions in lines 54-78.

2.1- While I understand that the authors have conducted a careful literature review, the conceptual structure of the scale and the rationale theory are somewhat unclear, and the explanations of the sub-concepts of the development scale appear to be disjointed.

-The recommendations of the 2.1 were added in the lines 69-78. Moreover, resource-based theory and human capital theory were mentioned in the introduction (lines 69-72) and discussion part (lines 603-607).

-Sub-concepts seem disjoint because we started each part with their definitions. However, we tried to connect sub-concepts by explaining their relations with remote working in the subsequent paragraphs as below:

• Cybersecurity (lines 164-166)

• Problem-solving (lines 186-194)

• Time management (lines 225-233)

• Verbal communication (lines 254-258)

• Written communication (lines 269-291)

2.2- Could you please explain a little more about the necessity of developing this scale? For example, in previous studies, is there a scale that can measure remote working skills or not? Does it exist globally or is there no Turkish version? If there is an existing remote working scale, what are the limitations of the existing scale?

-Recommendations of 2.2 were added in the lines of 54-68. Previous scales were competency, behavior, and attitude-oriented rather than skill-oriented. It is, of course, these scales are important contributions to remote working literature. However, we tried to bring a different approach by just focusing on skills. In light of this information, it could be said that the developed scale in this study is original and it can fill this gap in the literature. Therefore, there is no Turkish version as well as English or in other languages.

3- Methods：Missing are descriptions of the study design, inclusion and exclusion criteria for subjects, data collection methods, sample size calculation methods, and statistical methods for handling missing values.

3.1- I can't find a description of the research design. Is it a cross-sectional design? The type of study being used to test the properties (e.g., test-retest design, longitudinal study, cohort, cross sectional, etc.).

-Description of the research design was added as the second level new title in the 327th line. As stated in the lines between 330-331, the data is cross-sectional. 

3.2- Please state how the participants were chosen. Please explain the inclusion and exclusion criteria for the study subjects.

-As stated in line 333, the purposeful sampling technique was utilized in the study. Inclusion and exclusion criteria were added in the lines of 345-349. To sum up, inclusion/exclusion criteria were determined as having remote working experience. In addition, we used two control questions (“I can not use computer” and “I can not read and write”).

3.3- Data collection procedures are missing. An explicit description of how and when the participants were administered (e.g., in what setting) including data collection devices/system used (e.g. paper based, electronic administration) should be provided.

-The data collection procedures were added in the lines 349-358. The data were obtained electronically between April 2021 – February 2022.

3.4- Please describe sample size calculation.

-Sample size calculations of all samples were described in the lines 333-344. For the qualitative phase, we had 10 participants as Creswell and Creswell (2017) suggested. For the quantitative phase, we used the sample size criteria of “n > 65 and 40” for the content analysis. For the other analyses, we used the formula of Bartlett, Körtlik and Higgins (2001) to calculate sample size (n > 384).

3.5- The description of statistical methods in the Methods section is too brief. Also, although the statistical methods are described in the Results section, they should be clearly stated in the Methods section.

-The descriptions of statistical methods were moved to Methods from the Results section. Moreover, some additional information was given in each sub-title of Methods to improve the analyses. In this context all changes were made as below:

• Item generation and content validation (In lines 380-407)

• RWSS initial factor structure analysis (In lines of 457-469)

• RWSS factor construct verification and construct validity analysis (In the lines 474-481 and 492-503)

3.6- In particular, please add to the methods how the validity and reliability of the scale were determined and the statistical values set for the evaluation.

-The guidance of Hinkin (1998) for scale development was utilized in this study. Also, the steps of this guidance were put in order in the lines of 359-371. Additionally, analyses used for each step were stated along the same lines. Finally, a set of values for the evaluations was added as below: 

• Factor loadings (in lines 511-512). “ >,30” (Tabachnick and Fidell, 2007)

• CFA model fit indices (in lines 530-532). “χ2/df < 3, CFI > 0.90, TLI > 0,90, IFI > 0.90, 0.05 < SRMR < 0.08, 0.05 < RMSEA < 0.08” (Kline, 2011; Brown, 2006; Munro, 2005; Cheung and Rensvold, 2002).

• Cronbach’s Alpha (in lines 512-513). “ > 0,70” (Taber, 2018)

3.7- How did the authors handle missing values?

-There are no missing values in the study. Because all items in the online questionnaire were obligatory as stated in lines 351-352. 

4- Results: Descriptions of statistical methods in the Results section should be moved to the Methods section. I think the results should include the CFA Figures.

-All descriptions of statistical methods in the Results section were moved to the Methods section. CFA figure was added as Figure 1 in line 541.

5．Discussion: The discussion section is too short and does not include a summary of the study, interpretation of the results, the meaning and significance of the results of this study compared to previous studies, the novelty and generalizability of this scale.

5.1- The authors should compare the result to the criteria for good measurement properties (e.g., COSMIN criteria), and determine if the specific Measurement Property is sufficient or not.

-COSMIN (Consensus-Based Standards for the Selection of Health Measurement Instruments) is a measurement property mostly used in health sciences. Therefore, Hinkin's scale development guideline, which is widely used in social sciences, was taken as a reference in this study.

5.2- Generalizability issues related to the results should be discussed. For example, discuss if the results could be generalized to other populations given the sample studied.

-It is thought that the result of this study could be generalized to other populations in Türkiye. Because it was indicated in the materials and methods section, the sample of the study represented the population (total number of remote workers in Türkiye) via the formula of Bartlett, Körtlik and Higgins (2001). In addition, we administrated the questionnaires electronically so we were able to reach participants from country-wide.

-As we discussed in lines 668-679, the English version of the scale could be used globally. The scale can be used locally by adapting to other languages.

5.3- Although we have mentioned the development of the Turkish version as a limitation of this project, what are your thoughts on the development of an English version and the possibility of its use in other countries in the world?

-As we discussed in lines 668-679, it is thought that the English version of the scale can be applied globally. On the other hand, for researchers who wish to use the scale in their national language, the RWSS can be adapted by considering the cultural elements.

5.4- It would be desirable to state in the introduction and discussion whether remote work and workers' remote work skills in Turkey are common to other countries in the world, and whether there are any characteristics or differences among them compared to other countries in the world. 

-As we discussed in lines 668-679, it can be stated that Türkiye’s working conditions are at global standards and therefore the skills required in both face-to-face work and remote work are global.

---

## [Decision Letter · Decision Letter 1]

5 Feb 2024

Measuring remote working skills: Scale development and validation study

PONE-D-23-23682R1

Dear Dr. GÜNGÖR,

We’re pleased to inform you that your manuscript has been judged scientifically suitable for publication and will be formally accepted for publication once it meets all outstanding technical requirements.

Kind regards,

Dan-Cristian Dabija, PhD

Academic Editor

PLOS ONE

Additional Editor Comments (optional):

Thank you for implementing all suggestions and recommendations of the reviewers, which are now pleased with this version of the manuscript. Therefore I consider that the paper can be accepted.

Cristian Dabija

Reviewers' comments:

Reviewer's Responses to Questions

**Comments to the Author**

1. If the authors have adequately addressed your comments raised in a previous round of review and you feel that this manuscript is now acceptable for publication, you may indicate that here to bypass the “Comments to the Author” section, enter your conflict of interest statement in the “Confidential to Editor” section, and submit your "Accept" recommendation.

Reviewer #1: All comments have been addressed

Reviewer #2: All comments have been addressed

2. Is the manuscript technically sound, and do the data support the conclusions?

Reviewer #1: Yes

Reviewer #2: Yes

3. Has the statistical analysis been performed appropriately and rigorously? 

Reviewer #1: Yes

Reviewer #2: Yes

4. Have the authors made all data underlying the findings in their manuscript fully available?

Reviewer #1: Yes

Reviewer #2: Yes

5. Is the manuscript presented in an intelligible fashion and written in standard English?

Reviewer #1: No

Reviewer #2: Yes

6. Review Comments to the Author

Reviewer #1: I have read the referred article with keen interest. The information is interesting and innovative; conclusion section is interesting and authors can improve it further. I am recommending authors to do a little more work and add latest literate to support the study. The authors need to improve results section. The level of English is good and smooth, e.g., the language standard, specifically the grammar, of sufficient quality to meet scientific merit for publication. However, I suggest authors to double check for language quality. Describe scientific contribution of the study to the existing body of knowledge. I endorse this manuscript after minor revision as suggested. The topic is interesting and worthy of attention. The methodology is adequate and the conclusions are consistent with the reported data. The manuscript can be improved by expanding the references and citing some recently published articles on this topic.

Authors should consider the following recommendations:

- I recommend further improving the references by citing some of these recent studies on the topic:

Naeem, B., Aqeel, M., & de Almeida Santos, Z. (2021). Marital conflict, self-silencing, dissociation, and depression in married madrassa and non-madrassa women: a multilevel mediating model. Nature-Nurture Journal of Psychology, 1(2), 1-11.

Naeem, B., & Chaman, A. The Association of Adverse Self-Silencing and Marital Conflict with Symptoms of Depression and Dissociation in Married Madrassa and Non-Madrassa Women: A Cross-sectional Study.

Naeem, B. Nurturing the Soul: A Psychometric Analysis of the Spiritual Intelligence Inventory in Married Madrassa and Non-Madrassa Women.

Saif, J., Rohail, D. I., & Aqeel, M. (2021). Quality of Life, Coping Strategies, and Psychological Distress in Women with Primary and Secondary Infertility; A Mediating Model . Nature-Nurture Journal of Psychology, 1(1 SE-), 8–17.

Naeem, B., Aqeel, M., & de Almeida Santos, Z. (2021). Marital Conflict, Self-Silencing, Dissociation, and Depression in Married Madrassa and Non-Madrassa Women: A Multilevel Mediating Model. Nature-Nurture Journal of Psychology, 1(2), 1–11

Hafsa, S., Aqeel, M., & Shuja, K. H. (2021). The Moderating Role of Emotional Intelligence between Inter-Parental Conflicts and Loneliness in Male and Female Adolescents. Nature-Nurture Journal of Psychology, 1(1 SE-), 38–48

Rashid, A., Aqeel, M., Malik, D. B., & Salim, D. S. (2021). The Prevalence of Psychiatric Disorders in Breast Cancer Patients; A Cross-Sectional Study of Breast Cancer Patients Experience in Pakistan. Nature-Nurture Journal of Psychology, 1(1 SE-), 1–7. https://thenaturenurture.org/index.php/psychology/article/view/1

Sarfraz, R., Aqeel, M., Lactao, D. J., & Khan, D. S. (2021). Coping Strategies, Pain Severity, Pain Anxiety, Depression, Positive and Negative Affect in Osteoarthritis Patients; A Mediating and Moderating Model . Nature-Nurture Journal of Psychology, 1(1 SE-), 18–28. https://thenaturenurture.org/index.php/psychology/article/view/8

Aqeel, M., Nisar, H. H., Rehna, T., & Ahmed, A. (2021). Self-harm behaviour, psychopathological distress and suicidal ideation in normal and deliberate self-harm outpatient’s adults. Journal of the Pakistan Medical Association, 71(9), 2143-2147

Reviewer #2: Thank you again for the opportunity to review this interesting manuscript. I have found that the author has revised the manuscript addressing all the peer review comments.

7. PLOS authors have the option to publish the peer review history of their article (what does this mean?). If published, this will include your full peer review and any attached files.

Reviewer #1: **Yes: **Dr.Muhammad Aqeel

Reviewer #2: No

---

## [Editor Report · Acceptance letter]

1 Mar 2024

PONE-D-23-23682R1 

PLOS ONE

Dear Dr. GÜNGÖR, 

I'm pleased to inform you that your manuscript has been deemed suitable for publication in PLOS ONE. Congratulations! Your manuscript is now being handed over to our production team.

Kind regards, 

on behalf of

Professor Dan-Cristian Dabija 

Academic Editor

PLOS ONE